# New Insights into Plant TPK Ion Channel Evolution

**DOI:** 10.3390/plants10112328

**Published:** 2021-10-28

**Authors:** Siarhei A. Dabravolski, Stanislav V. Isayenkov

**Affiliations:** 1Department of Clinical Diagnostics, Vitebsk State Academy of Veterinary Medicine [UO VGAVM], 21002 Vitebsk, Belarus; SergeDobrowolski@gmail.com; 2International Research Center for Environmental Membrane Biology, Foshan University, Foshan 528000, China; 3Department of Plant Food Products and Biofortification, Institute of Food Biotechnology and Genomics NAS of Ukraine, 04123 Kyiv, Ukraine

**Keywords:** KCO3, TPK, two pore potassium channel, molecular phylogeny, molecular evolution

## Abstract

Potassium (K) is a crucial element of plant nutrition, involved in many physiological and molecular processes. K^+^ membrane transporters are playing a pivotal role in K^+^ transport and tissue distribution as well as in various plant stress responses and developmental processes. Two-pore K^+^-channels (TPKs) are essential to maintain plant K^+^ homeostasis and are mainly involved in potassium transport from the vacuoles to the cytosol. Besides vacuolar specialization, some TPK members display different membrane localization including plasma membrane, protein storage vacuole membrane, and probably the organelles. In this manuscript, we elucidate the evolution of the voltage-independent TPK (two-pore K^+^-channels) family, which could be represented in some species by one pore, K^+^-inward rectifier (Kir)-like channels. A comprehensive investigation of existing databases and application of modern bioinformatic tools allowed us to make a detailed phylogenetic inventory of TPK/KCO3 (KCO: potassium channel, outward rectifying) channels through many taxa and gain insight into the evolutionary origin of TPK family proteins. Our results reveal the fundamental evolutional difference between the first and second pores, traced throughout multiple taxa variations in the ion selection filter motif, presence of thansposon, and methylation site in the proximity of some KCO members and suggest virus-mediated horizontal transfer of a KCO3-like ancestor by viruses. Additionally, we suggest several interconnected hypotheses to explain the obtained results and provide a theoretical background for future experimental validation.

## 1. Introduction

Potassium (K) is one of the essential nutrients required for plant growth and development. This mineral is the most abundant cation of plants and K^+^ is crucial for a multitude of physiological processes such as protein translation, enzymatic catalysis, screening of negative charges, and providing turgor [1,2,3]. Cytosolic K^+^ also plays an important role in plant adaptive responses to environmental stresses such as drought and salinity [4,5,6] and plant disease [6,7]. 

Plant tissue K^+^ content is typically in the order of ~100 mM on a fresh weight basis and sustaining this level requires a sophisticated array of membrane transporters. K^+^ transport is mediated by K^+^ transporters that include active K^+^ carriers and passive K^+^ channels. The former can be classified as belonging to the CPA, Trk/Ktr/HKT, or KT/HAK/KUP families [8,9] whereas K^+^ channels are encoded by members of the voltage-gated Shaker channel family or the voltage-independent two-pore K^+^-channel (TPK) family, which in some species includes a one pore, K^+^-inward rectifier (Kir)-like channel named KCO3 in Arabidopsis (Figure 1) [10,11,12]. 

TPKs activity is insensitive to membrane voltage. However, many TPK channels contain one or two Ca^2+^ binding EF-hands in the C-terminus and channel activity has been shown to depend on cytoplasmic Ca^2+^ levels [13,14,15,16]. TPKs can have domains for the binding of 14-3-3 proteins in their N terminus. Phosphorylation of this domain by kinases such as KIN7 [17] and subsequent 14-3-3 binding greatly affects TPK activity [17,18,19]. Cytoplasmic pH is another factor impacting channel open probability [14] as is interaction with various kinase isoforms of the CIPK (CBL-interacting kinase) family [20,21]. Furthermore, TPK channels can alter activity upon membrane stretching or the creation of trans-tonoplast osmotic gradients, indicating a role in intracellular osmosensing [22]. 

Extensive studies have shown that the majority of TPKs and KCO3 is targeted to the tonoplast (i.e., the membrane of the lytic and/or storage vacuole) [14,18,23,24,25,26,27]. However, isoforms have also been detected in the plasma membrane (e.g., AtTPK4) and possibly the thylakoid membrane [28]. However, the questions of AtTPK3 thylakoid localization as well as physiological function remain open and need further experimental support [29]. Functional TPK channels contain four pore domains (e.g., [12,30]) and the application of FRET and BiFC techniques revealed the existence of homodimeric forms of AtTPK1 and AtTPK5 [17,23]. In addition, the formation of heterodimers between AtTPK1 and AtTPK3 has been suggested [31]. Formation of stable dimers of AtKCO3 also occurs [32], but not surprisingly, did not show functionality. TPK channels are probably involved in the maintenance of K^+^ homeostasis in plant cells by controlled intracellular K^+^ transport from and into organelles, particularly the vacuoles [14,33]. Furthermore, a recent study on *A. thaliana* suggests that TPKs and TPC1 channels may interact at the tonoplast to endow excitability to this membrane [31]. Nevertheless, functional aspects of many TPK/KCO channels remain largely unexplored.

The panoply of regulatory mechanisms, membrane localizations, and potential functions impinges on the intriguing question of how various KCO and TPK isoforms relate to each other in an evolutionary sense. Phylogenetic studies of TPK/KCO channels have been conducted in the past decades [19,25,34], but were based on a limited number of sequences and species. Recent progress in the genome sequencing of plant species from various taxonomic groups as well as data from other taxa has greatly facilitated comparative studies across a much wider variety of species. In this study, we compiled a comprehensive TPK/KCO inventory and carried out phylogenetic and structural analyses on both the first and second pore domains of the TPK channels. Our detailed domain architecture analysis suggests that the two-pore domains of plant TPKs have a distinct evolutionary antecedent. The first pore exhibits a highly conserved sequence in the pore loop and a TxGYGD selectivity filter analogous to that found in archeal KcsA channels and subsequently in Shaker type voltage-gated (VG) channels. The second pore has an altered selectivity filter (TxGFGD) and is more likely to have ancestry based on a KCO lineage, since our phylogenetic analyses showed the occurrence of KCO type channels in a far wider number of species than hitherto reported. Additionally, obtained results suggest virus-mediated transfer of KCO-like ancestral proteins from several potential hosts. 

## 2. Results

### 2.1. Analysis of the Arabidopsis C- and N-Terminal Pore Regions

TPK and KCO type channels contain two and one ‘cation channel domain’ (pFAM PF07885), respectively (Figure 1), consisting of two transmembrane helices, a pore loop, and a selectivity filter. Four of these ‘pore module’ oligomerises represent the minimal structure of a functioning K channel. A multiple sequence alignment of Arabidopsis TPK/KCO channels revealed the existence of some important differences in modular structure between different family members: first, AtKCO3 is missing the N-terminal pore module; second, AtTPK4 has deletions outside of the pore modules in both N- and C-terminal domains that led to the removal of potential regulatory sequences such as the 14-3-3 binding motif and EF-hands; and third, the sequence of AtTPK4 was the most different from the consensus sequence of the entire TPK family (not counting AtKCO3) (Appendix A).

Based on a very limited distribution of KCOs in plants and similarity, current evolutionary models assume that KCO3 is the product of a relatively late TPK4 gene duplication event [19,25,34]. To further elucidate phylogenetic relationships within the Arabidopsis TPK/KCO family, we separately analyzed the N- and C-terminal pore modules in terms of the sequence (Figure 1 and Appendix A), sequence similarity (Appendix A), and phylogeny (Figure 2). It was found that all N-terminal pore regions were closely related, while the C-terminal copies were more divergent, suggesting ongoing evolutional processes on that domain (Figure 3). The C-terminal pore region of AtTPK4 was the most different from that of other TPK isoforms (Appendix A). Furthermore, pore domain comparison confirmed our alignment results (Appendix A), showing that AtKCO3 lacks the N-terminal pore domain as was previously shown [34,35].

### 2.2. Pore Alignment and Analysis of the Voltage-Dependent Potassium Channel Signature

To test whether the observed differences between the first and second pores of Arabidopsis TPK family members are present in other species, the sequences of the TPK proteins from different taxa (photosynthetic bacteria, green and red algae, “low” plants, mono and di-cotyledons, SAR, metazoa, and viruses) were analyzed. Each pore sequence was extracted and examined individually. This showed that some pores of the TPK proteins exhibited significant similarity to the pores of voltage-dependent (PVG) channels such as AKT K^+^ channels, represented by the ion transport protein domain PF00520 (Pfam database classification) (Appendix A) (E-value (≤0.001). Interestingly, both domains (‘ion channel’ PF07885 and ‘ion transport protein’ PF00520) belong to the voltage-dependent potassium channel superfamily (PLN03192). The similarity was especially evident in the region of the selectivity filter GYGD motif and could point to a common evolutionary origin [36]. However, the studied taxa differed greatly where the presence of such ‘voltage signatures’ (V-signatures) is concerned: Archaea, Bacteria, and SAR, organisms that predominantly have K^+^ channels with a single ion channel domain (i.e., only one pore loop) typically show V-signatures. The metazoa (multicellular animals), on the other hand, have mainly two-pore channels, neither of which shows a V-signature. Streptophyta (higher plants) have a distinctly different configuration with a V-signature in the first pore but not the second. Green and red algae have several intermediate patterns, where two-pore domain proteins show none, one, or two V-signatures. Amino acids, matching VDC signatures of the model species (e.g., Arabidopsis, rice, and *Physcomitrella patens*) are highlighted on the alignments (Appendix A). The recent structural and evolutional revision of plant VG K^+^ channels suggests that these proteins form a completely different (from metazoan) channel group and originated from another (not metazoan) procaryotic ancestor [37]. 

### 2.3. Variation in the Selectivity Filter Motif in the First and Second Pore

Ion selectivity is defined by the GYGD motif, where Y and D residues are crucial for K^+^ selectivity and open state, respectively [38]. We found the exact match to the GYGD motif of TPK sequences from photosynthetic bacteria to high plants (Appendix A). However, for bacteria and algae, we noticed a high degree of motif variation, where Y is often replaced by L or F. According to this observation, it could be suggested that such amino acid substitutions in the GYGD motif reduce the selectivity strength and provide the ability to transport other monovalent cations, most probably Na^+^ [38]. The second crucial amino acid in the GYGD motif is D, which is a crucial determinant of open state stability [38]. It was found that the first pore of KCO-like from photosynthetic bacteria KCO-like exhibited the variability of both sites, while for the second pore selectivity filter motif, it looks like GxxD (Appendix A). Unfortunately, for photosynthetic bacteria, there is only a limited number of sequences without VDC available, therefore, we cannot state that this conclusion would be the ultimate. In contrast with photosynthetic bacteria, the TPKs of green and red algae had variations only for Y (to F and L) in the first pore, but for the second pore, we found different variants from conserved GYGD to the complete absence of this motif (Appendix A). 

A rather interesting example of the motif variation was found in low plants and monocotyledons (Appendix A). While the first pore GYGD was mostly conserved among these taxonomic plant groups, the second pore was conserved, but also not well-aligned/shifted. The functional outcome of such disturbance remains unclear. In the majority of Dicotyledons, in contrast, both pores have conserved a GYGD motif, with the only exception for a *Glycine max* (A0A0R0GFP1_N) for the first pore, and KCO3 with GFGD in *Arabidopsis thaliana* (Appendix A). However, it was reported that one of the Tobacco TPK isoforms may have variations (VHG and GHG instead of the GYG) in the second pore selectivity filter [39].

### 2.4. TPK/KCO Phylogeny

#### 2.4.1. General TPK Phylogeny (Based on the Alignment of Full-Length Proteins) 

An initial phylogenetic analysis was carried out using the ‘ion channel’ domain PF07885, which consists of two transmembrane domains (TMDs) interspersed by a pore loop and henceforth referred to as a ‘pore region’ (Figure 1). This analysis showed the presence of TPKs and KCOs in a wide range of taxa across all biological kingdoms, which included many different plant species (Figure 4, Appendix A). Within plant species, the TPKs/KCOs cluster into two main clades, which can be further sub-divided into sub-clades ‘A’ and ‘B’. Clade I contains the evolutionary most recent TPK members found in Embryophyta and flowering plants (mono- and dicotyledonous). Within this clade, sub-clade A contains the more ‘advanced’ TPKs from higher plants, while sub-clade B contains members from early land colonizers such as *Physcomitrella patens*, together with *Marchantia polymorpha* and *Selaginella moellendorffii*. 

Clade IIA contains TPK members from only monocotyledon species, with high similarity to AtTPK1, whereas clade IIB forms a collection of TPK channels from dicotyledons and more primitive flowering plants such as *Nelumbo nucifera* and *Cinnamomum micranthum*. Interestingly, the branch of the freshwater alga *Klebsormidium nitens* and bryophyte *Physcomitrella patens* acts as a parental form for the entire clade II. The lack of relationship between AtTPK4 and any of the other Arabidopsis TPKs led to this channel being positioned apart from other TPK isoforms, between clade I and II.

Previous work reported the presence of a KCO (AtKCO3) only in *Arabidopsis thaliana* and *A. lyrata* [19,34], positioned near AtTPK2. However, our analysis showed that AtKCO3 is located on a separate branch, together with KCO3-like proteins from the lower plant *Chara braunii*, the lycophyte *Selaginella moellendorffii*, apple (*Malus domestica*) and cucumber (*Cucumis sativus*). Thus, these data not only show that KCO type channels are far more ubiquitous than previously reported, and their presence in the *S. moellendorffii* genome supports the idea that KCO-like single-domain ion channels could be the basis for all TPK K^+^ channels since lycophytes form one of the oldest lineages of extant vascular plants. 

#### 2.4.2. Domain-Based TPK Phylogeny (First and Second Pores Taken Separately)

The phylogeny analysis was carried out with the N- and C-terminal ion channel domains PF07885 (Figure 1), which were further referred to as the first and second pores, respectively. We made a major observation, that, in general, TPK proteins with a VDC signature (from the AKT K^+^ channels, representing ion transport protein domain (PF00520)) and VDC-free proteins (depicted as NV) tend to form separated clusters (Appendix A). We found sub-clusters formed separately by VDC and VDC-free proteins for every taxon: mono- and dicotyledons, green and red algae, viruses, bacteria, and archaea. Some minor exclusions for this rule could be found in viral sub-clades: Clade Ia had one exception, while Clade Ib was represented rather by a mix of VDC and VDC-free proteins. Clades Ic and Id, in contrast, were formed by VDC and VDC-free proteins, respectively. Interestingly, land plants have formed separated sub-clusters: one for VDC-free (Clade IIa) and two for VDC (Clade IIb and IIc) proteins. Other taxa, in contrast, were clustered in many small groups of closely related proteins (red and green algae, archaea, bacteria, metazoa, and SAR). 

As a possible site of an ancient gene transfer (or convergent evolution), we noticed several separated branches, where proteins from different taxa were clustered together. Branch 1, where Archaea VDC-free protein A0A133VJR6 was clustered together with proteins from photosynthetic bacteria *Kouleothrix aurantiaca* (VDC-free A0A0P9D655) and *Anabaena* sp. WA102 (VDC-containing A0A0K2LYX4). The second branch contains two VDC proteins: archaeal A0A062V1H8 and bacterial *Bilophila wadsworthia* (E5YBI1). Similarly, the third branch contains proteins from bacteria and archaea: two VDC archaeal proteins (*Heimdallarchaeota archaeon* A0A2U3CBL9 and *Halopelagius longus* A0A1H1DI30) and two bacterial (VDC protein *Vibrio nigripulchritudo* U4KGM9 and VDC-free *Sedimentisphaera salicampi* A0A1W6LMM0). 

Previously conducted TPK channel phylogeny analyses have had some limitations. For example, Voelker et al. (2010) [19] only used 12 plant species (“low” plants, mono- and dicotyledons); Gomez-Porras et al. (2012) [34] only used five plant species; and Marcel et al. (2010) [25] used much wider taxa (68 channels in sum) also including protist, fungi, and multicellular animals. However, our analysis (Appendix A) had some advantages: (1) analysis of domains (not full-length sequences) allowed us to also include single-pore channels (KCO3-like proteins) from different species (plants, bacteria, archaea, viruses); (2) separated analysis of first and second pores (“N” stands for N-terminal and “C” for C-terminal) allowed us to show the different origin of first and second pores; (3) channels from all available taxa (metazoa, fungi, archaea, bacteria, virus, green and red algae, bryophyta, mono- and dicotyledons, SAR) allowed us to identify possible points of horizontal gene transfer (Branches 1 to 3) between different taxa; and (4) highlighted the absence of a VDC signature (“NV” stands for “no voltage gate”), which allowed us to trace the evolution pathway of different pores, because in most cases, “NV” is combined with “C”. Details of the used legend and abbreviations used are also presented in Appendix A. 

In total, these results support our assumption on the different origins of the first and second pore. The close inter-taxa connection was found only between Archaea and bacteria. Land plants have formed separated clusters, while proteins from other taxa were represented by many small branches of closely related proteins. 

## 3. Discussion

The results of our structural and phylogenetic analyses of TPK proteins reveals the significant complexity of these channels and their evolution. Our study provides new evolutional and structural insights into plant TPK/KCO3 proteins. Furthermore, comparative and evolutionary analysis of plant TPKs/KCO3 with other taxa revealed substantial structural differences of plant TPKs. Interestingly, we found the presence of VDC signatures only in the first pores for TPKs from higher plants. Further application of bioinformatic structural analysis revealed domination of TPK split forms (first pore with VDC signature and second VDC free) and strict GYGD selectivity filters in higher plants. Perhaps, such structural features of TPKs became beneficial for the higher plants after land colonization, and can be connected with vacuolar specialization and possible involvement in cellular signaling in the form of Ca^2+^ vacuolar content control and potassium homeostasis. In addition, phylogenetic analysis of Arabidopsis KCO3 suggests that KCO-type channels may be an ancestral form to all TPKs. The presence of this type of channel in genomes of some higher plant species could be explained by potential viral transfer from some protozoan species to the ancient algal endosymbiont. Perhaps the KCO3-like channels in higher plants may interact with “modern” TPKs and regulate their activity. Taking together the literature analysis and the results of our study, we can suggest the possibility for an alternative evolutionary pathway of KCO3-like channels in plants that is different to the dominated hypothesis of gene duplication and subsequent pore deletion. 

Results of our general phylogenetic analysis indicate that TPKs/KCOs proteins are distributed between two main clades, which can be further sub-divided into sub-clades ‘A’ and ‘B’. Albeit some similarities with previous phylogenetic reports [19,25,34], our new phylogenetic tree comprises the extended number of subclades and a large array of different plant species and forms of TPKs/KCO channels. Thus, the current phylogenetic inventory is the most updated reconstruction of plant TPK phylogeny, supporting the idea of the multiplication and complexation of TPK isoforms during land colonization and evolution in general. Taken together, our results indicate that plant TPK channels are considerably different from their counterparts from other taxa and may have several possible options of evolutional and structural development. Results of our domain-based phylogeny analysis suggest the different origins of the plant TPK first and second pores. The first pore contains the VDC signature (the dominant feature for all taxa, but metazoa); the second pore without a VDC signature, was, most probably transferred from some metazoa species via viruses to the ancestors of the green lineage (Figure 5). 

Furthermore, new questions were raised during our study. To answer the questions and explain the results of our study, several interconnected hypotheses and possible ways to check them were suggested: 

(1) Origin of voltage-dependent channel signature (VDC signature). Based on the presence of VDC and GYGD selectivity motif in the majority of pores, we hypothesized that TPK channels have originated from ancient voltage-gated/dependent channels. The presence of VDC signatures only in the first pores could suggest the importance of this domain and first pore for appropriate functioning of TPK. Indeed, based on Hamamoto’s results [39], they proved the absolute significance of the GYGD motif in the first pore, while it was less important in the second pore, we could explain wider variation in the second pore and high degree of conservation for the first pore. Possibly, the further evolutionary development of TPK channels in plants is tightly connected with vacuolar specialization. For example, a recent study showed that TPK channels play an essential role in voltage-stabilizing of the vacuolar membrane as a security valve [31]. Such an important service of TPKs to maintain appropriate membrane voltage homeostasis of vacuoles may require the presence of a VDC domain for interaction with other vacuolar voltage-dependent channels such as TPC1 [31]. In contrast with a range of different types of vacuoles in plants, animal cells have lysosomes as analogues of the plant lytic vacuoles. Besides lytic functions, plant vacuoles are important storage compartments for the ions and have to be maintained and controlled by far more sophisticated transporter machinery. Therefore, perhaps, animal TPK counterparts-tandem pore (TWIK-like) channels do not need VDC motifs to perform their physiological roles. Another option to have VDC motifs for the plant TPK is probably an evolutionary development from the ancestral voltage-gated channel. For example, it was reported that plant VG K^+^ channels originated from a completely different metazoa prokaryotic ancestor [37]. Perhaps the VDC motif pore of plant TPK channels is derived from an archaic VG K^+^ channel. However, the physiological significance of the VDC-mediated regulation and the primary role of the first pore in the ion transfer remain unknown and require further experimental investigation. As a possible experimental approach, we could suggest introducing point mutations into the second pore sequence, imitating the VDC-signature. The suggested approach will help to answer some of the following questions:

1. How such a modification would influence the regulation of the second pore and the entire channel in general?

2. Would it change the interaction with KCO3 or assemble into the dime/tetramer structure?

In addition, the next interesting approach would be to exchange the second pore with another first pore with VDC to create a channel with two pores with VDC. We believe that the recruitment of these experimental approaches will help to clarify the question of plant TPK origin.

(2) Variation and selectivity of GYGD filters. Our analysis of GYGD filters in different TPKs led to the appearance of the main question: What is the relationship between the absence of the VDC signature in the second pore; variation in the second pore GYGD selectivity filter motif, and secondary role of the second pore in the ion transfer? Because proteins could interact and affect each other [35], the second pore could regulate the activity/function/selectivity of the first pore. Furthermore, the recent data suggest that Arabidopsis single pore KCO3 components are required for the formation of a functional K^+^ channel and oligomerization [35]. Thus, it would be very interesting to evaluate the role of Arabidopsis KCO3 in the regulation of TPK channels. However, not many plant species have a single-domain/pore KCO-like homologue similar to Arabidopsis KCO3. Thus, some other mechanisms could be involved to perform a similar function. For example, it was postulated that AtTPK1 and AtTPK3 can interact with each other and form functional heterodimers [31]. It is possible that the function of the shifted GYGD motif in the second pore of low plants and monocotyledons would play such a role. Therefore, a wider investigation is required to match species with shifts in the GYGD motif in the second pore and species without KCO3 homologues.

Furthermore, it could be another possible mechanism of TPK regulation such as modifications in the GYGD motif (Y → F and L) as it is used in simple organisms and low plants. In addition, such shifts of GYGD motifs in the TPK structure of monocots could be linked to the different vacuole specialization and pH. Differences in the compositions of vacuolar membranes as well as in pH of the vacuolar lumen may require different conformation states and perhaps regulations. For example, it was shown that two TPK isoforms from rice and one from barley could be targeted to the different types of vacuoles [27]. However, this hypothesis requires experimental confirmation. As an interesting experimental setup, we suggest thee inversion of one or both pores (N-to-C direction). How such modifications would influence the direction of K^+^ transport, di/tetramerization, and selection of the type of target membranes will help to clarify the mechanisms of TPK regulation and selectivity. 

(3) Roles of TPKs in plant evolutional adaptation to the different environments. As a line of the general evolution of the TPK protein family, we found that simple organisms have several proteins with diverse substitutions of the Y and D positions in the selectivity filter of pore regions, potentially required to provide resilience in the control of the membrane potential. These amino acid substitutions allow use of a limited number of proteins for diverse application, adaptation, and keep their genome small at the same time. We could only speculate, that land colonization by higher plants was one of the driving forces to develop strict ion specialization of this type of channel due to the higher environmental complexity of terrestrial habitats. Land plants have developed specialized transporters to maintain Na^+^ and K^+^ balance independently, thus TPK channels have shifted specificity to K^+^, while other channels are employed for Na+ or other cation transport. Perhaps, ion (K^+^) specialization of TPK channels could provide stricter control of the membrane hyperpolarization and protect cells from potentially toxic cations. As was shown recently, the activity of the vacuolar two-pore Ca^2+^-channel 1 (TPC1) channel depends on the presence of K^+^ in the luminal side of vacuoles, whereas elevated vacuolar Na^+^ levels prevent activation of the TPC1channel [31,40]. Thus, besides maintenance of K^+^ homeostasis, acquisition of K^+^ specificity for TPKs could be an evolutionary advantage to control Ca^2+^ luminal level and be involved in the processes of intracellular signaling. 

In addition, a wider specialization of the bacterial/algal TPKs was required as a way to adapt to severe environmental conditions. Higher plants, in contrast, inhabit soil, where challenging factors are different, thus, require new adaptation features. Additionally, plants have developed another protection shield, the cell wall, making them less sensitive to the changes in environmental conditions. 

(4) Role and origin of KCO3. Interestingly, the Arabidopsis single-pore/domain KCO3 GFGD motif is very similar to bacterial KCO3-like type GFGD, thus, suggesting its ancient nature (Appendix A). Due to the presence of methylation and transposon in the proximity in the Arabidopsis genome, we could suggest that this gene transfer was possibly mediated by a virus from metazoa (Appendix A). This hypothesis is partially supported by the dominance of the VDC-free TPKs in the metazoa and several viruses, with defined VDC-free TPKs, which could serve as transfer agents. As an experimental approach to check this hypothesis, we could suggest the introduction of the bacterial KCO3-homologue into Arabidopsis (KCO3 KO). In addition, if this hypothesis is true, the introduction of such bacterial protein in WT Arabidopsis would have a similar phenotype to the KCO3 overexpressing plant. Another interesting experiment would be to introduce first pore to the KCO3 (with VDC signature). Localization preferences and physiological functions of such protein would be interesting to compare to native KCO3 and other TPKs.

Surprisingly, our search for the ion channel domains has also revealed the presence of the homologues of this domain in several virus taxa (in total, 87 proteins, Appendix A) such as Chlorovirus and Prasinovirus, which are known to affect green algae. Numerous *Paramecium bursaria* and *Chlorella* virus strains are particularly interesting because they are known to invade the Chlorella endosymbiont of the protozoan *Paramecium bursaria*, suggesting endosymbiotic gene transfer from the protozoan organism to the green algae. Furthermore, a recent comprehensive bioinformatics analysis [41] suggests that up to 10% of the algae genomes are formed by endosymbiotic gene transfer by giant viruses and supports this assumption.

However, it could just be another possible option of KCO3 origin. For example, only the second pore was transferred by the virus and at some point, fused with a native VDC-containing plant TPK channel, resulting in the current split form with VDC containing signature and VDC-free pores. This hypothesis could explain the presence of both proteins with a VDC signature and VDC-free (Appendix A). Thus, KCO3 most probably, as an original form of TPK ion channel, is delivered to the ancient green algae (or red and green proto ancestor) from some protozoan species [42]. Nonetheless, the suggested hypothesis needs further experimental approval.

(5) General evolution of TPKs. According to our data, the evolution of plant TPKs can have a more complex picture than previously thought. Alternatively, TPK pores could have different origins, so the first pore could originate from voltage-gated/dependent K^+^ channels, while the second from another type of monovalent ion channel. However, this hypothesis about the origin of both pores also needs to be revealed further by experimental and structural studies. Their interaction and co-regulation on the molecular and structural levels would help to answer these questions. Some recent papers have suggested several whole-genome duplication events to explain the number and redundancy of TPKs in plants, while the absence of the first pore of KCO3 can be explained by the partial deletion [34]. Based on our analysis of the pore/domain distribution (Appendix A), we could conclude that a single domain/pore structure is usual for Archaea, bacteria, and SAR (either with VDC or without, but not both) with the prevalence of pores with the VDC signature. In contrast, Green and Red algae usually have 2 pores/domains of the same type structure (both with or both without VDC signature). The split form of TPKs (first pore with VDC signature and second without) are underrepresented in algae. However, this TPK structural form sharply dominates in land plants, suggesting significant pressure on these genes/function, or acquisition of this form (from virus?) (Figure 5). Thus, it is very likely that the quick spread of TPK split forms (first pore with VDC signature and second without) among land plants is a very effective and useful adaptation in new environments, allowing them to be more resilient and flexible. 

Keeping in mind the discussed results and described limitations, we have to admit that the analysis of full-length proteins could not provide any new information about TPK family evolution. Surely, clade distribution and tree topology could slightly vary from paper to paper (depending on phylogeny reconstruction method, used sequences, and bioinformatic tool). Thus, reconstruction of the TPK phylogeny based only on plant sequences (Appendix A) does not provide any answers to the raised questions. Similarly, domain-separated tree reconstruction has only confirmed that C- and N-terminal domains form separated clusters and have different evolution origins (Appendix A). For these reasons, we used several sequences from every high-ranked taxa (Appendix A). Such an approach allowed us to define the possible evolutional origin of TPK and KCO proteins, and suggest several possible points to horizontal gene transfer (between bacteria and archaea, and green photosynthetic bacteria and archaea) and possible routes for virus-mediated gene transfer (Appendix A). However, a limited number of used sequences could also be one limitation of the study. Future research would require a much wider species representation for every high-ranked taxa to provide more definite points of HGT and routes for virus-mediated gene transfer.

The analysis of many sequences representing many taxa has been successfully used to explore the evolution of many gene families and to define small (limited number of genes were transferred) and large scale (a significant part of the genome was transferred) HGT and virus-mediated gene transfer [43,44,45].

## 4. Materials and Methods

### 4.1. Identification of the TPK Family Members

Members of the TPK family (ion channel domain PF07885) were identified with searches in the NCBI (National Center for Biotechnology Information), InterPro 85 [46], and Pfam 34.0 [47] databases. All partial and fragmented sequences were eliminated. The presence of the ion channel domain (PF07885) and matched to partial voltage-dependent potassium channel (PLN03192) (further referred to as VDC signature) were checked with CD-search v3.19 (NCBI) [48] and MOTIF search (KEGG 98) [49] tools with E-value (≤0.001). Domains, fused to the nitroreductase domain, were verified with the same tools and threshold.

### 4.2. Multiple Sequence Alignments and Phylogenetic Analysis

Ion channel domain sequences were extracted from the Uniprot database (The UniProt [47]; for proteins with two ion channel domains, the sequences were marked as N and C-terminal, respectively) and multiple sequence alignments were performed with MUSCLE [50]. Testing of substitution models and phylogenetic analysis was carried out using the MEGA X software (default parameters) [51]. For the maximum likelihood tree [52], the LG [53] and WAG substitution models were selected, assuming an estimated proportion of invariant sites and four gamma-distributed rate categories to account for rate heterogeneity across sites. The JTT substitution model [54] was used for reconstruction with the neighbor-joining [55] method. The gamma shape parameter was estimated directly from the data. Reliability for the internal branch was assessed using the bootstrapping method (1000 bootstrap replicates).

Multiple sequence alignments to depict VDC signature and GYGD motif, TM (transmembrane), EF-hand, and 14-3-3 domains were performed using MUSCLE [5] with default settings in Ugene 38 software [56] and colored as % of identity.

## 5. Conclusions

In total, our results indicate several new hallmarks of the TPKs’ evolution: (1) presence of the VDC signature; (2) evolutionally conserved variations in the ion selection motif; (3) sights of the viral transfer; and (4) fundamental evolutional difference between the first and second pores. Furthermore, during the analysis of our results, several different but interconnected hypotheses were suggested to explain observed differences in structure, selectivity, and evolutional development of plant TPKs. The possible ways to approve or disapprove these hypotheses were discussed. Thus, to better understand the evolution of TPK channels, their role in plant physiology, and molecular basis of their activity, further appropriate experiments will help to solve the questions raised in our study.

## Figures and Tables

**Figure 1 plants-10-02328-f001:**
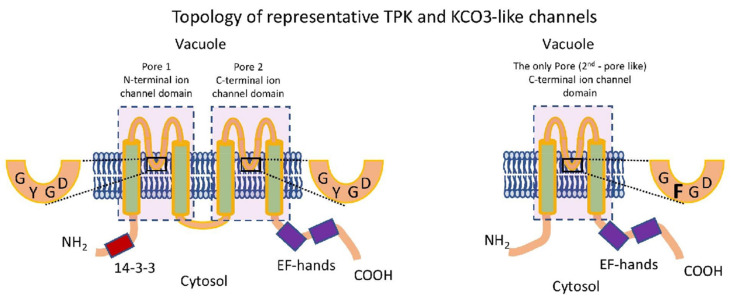
Representative topology of TPK and KCO3 channels. A tandem arrangement of two ‘ion channel’ domains (PF07885) (for TPK) and a single ion channel domain (for KCO3) is presented. Each domain (rectangle) contains two transmembrane helices, an intermediate pore loop, and a selectivity filter. KCO type channels lack the N-terminal domain.

**Figure 2 plants-10-02328-f002:**
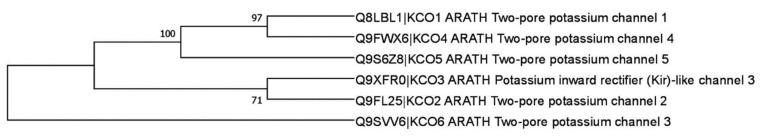
Phylogenetic tree of the TPK and KCO proteins of *Arabidopsis thaliana*. Full-length proteins from Arabidopsis were used for phylogeny reconstruction with the maximum likelihood method and WAG substitution model.

**Figure 3 plants-10-02328-f003:**
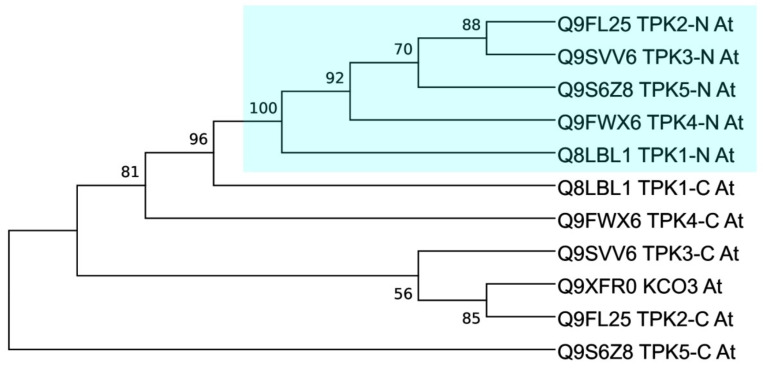
Phylogeny estimation of the *Arabidopsis thaliana* N- and C-terminal pore modules. The maximum likelihood method and LG substitution model were used. N-terminal pores are highlighted.

**Figure 4 plants-10-02328-f004:**
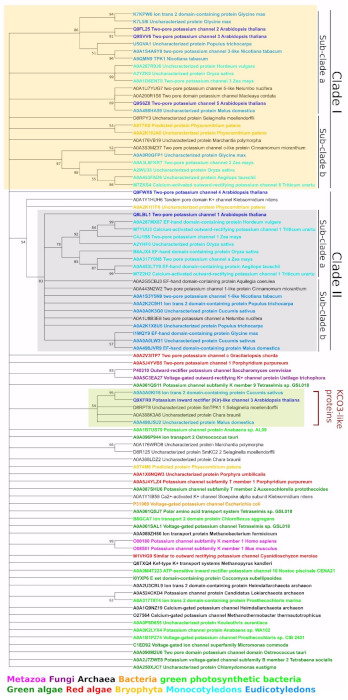
Phylogeny estimation of the TPK proteins. Full-length proteins from selected taxa were used for phylogeny reconstruction with the maximum likelihood method and LG substitution model.

**Figure 5 plants-10-02328-f005:**
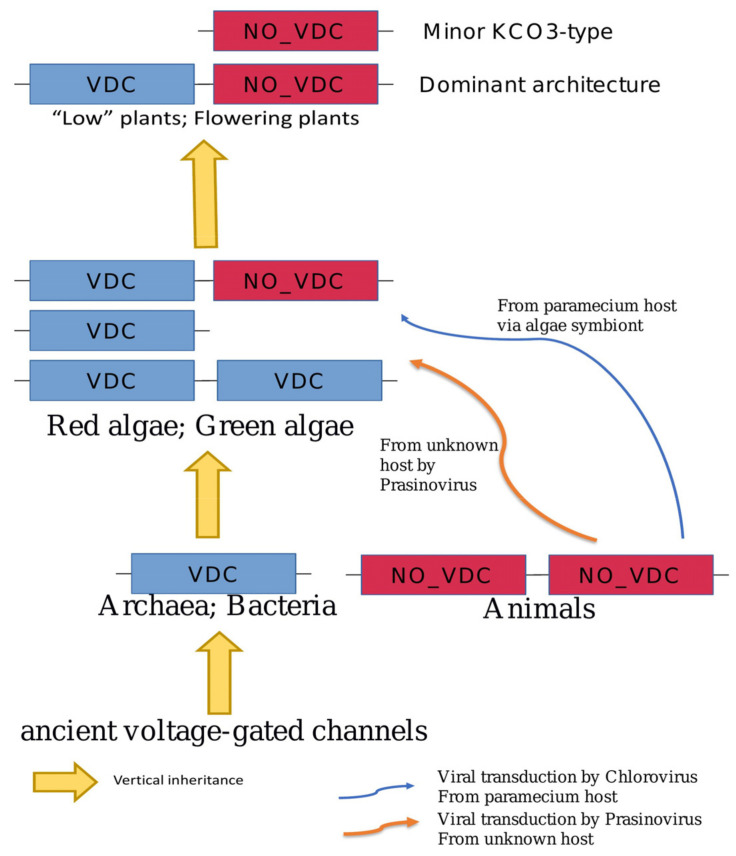
Simplified scheme of possible evolutional pathways of the TPK channels.

## Data Availability

All presented data are available upon personal request.

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
