# Peer review of "New Insights into Plant TPK Ion Channel Evolution"

_plants, 2021, doi:10.3390/plants10112328_

Round 1

Reviewer 1 Report

The work presented by the authors is well planned, they try to answer to relevant questions about evolution and I think that the work is adding relevant information that can help in the study of the mechanisms involved in the regulation of ion exchanges.

I don't have a great experience in phylogenetic analyses, so I cannot express a consideration about the software and methods used in this work. However, I've been looking at similar works already published and I see that the authors have followed a general approach that seems to be  accepted.

In my opinion, this work fullfill the objectives of this journal and I think that the information presented is clear and complete, so that it makes  it suitable for publication in the current state.

Author Response

Dear Editor and Reviewers,
We greatly appreciate your critical evaluation of our manuscript and helpful comments. Our reply to your comments would be provided point by point, where “A” stands for “Authors”, and “L” for “Lines”, where changes have been implemented. The language of the entire manuscript has been checked and corrected.

____________________________________________________________________________

The work presented by the authors is well planned, they try to answer to relevant questions about evolution and I think that the work is adding relevant information that can help in the study of the mechanisms involved in the regulation of ion exchanges.

I don't have a great experience in phylogenetic analyses, so I cannot express a consideration about the software and methods used in this work. However, I've been looking at similar works already published and I see that the authors have followed a general approach that seems to be accepted.

In my opinion, this work fullfill the objectives of this journal and I think that the information presented is clear and complete, so that it makes it suitable for publication in the current state.

A: We thank you for your valuable opinion regarding our work.

Reviewer 2 Report

The article have good collection of information on TPK ion channels, especially on its evolution. It is interesting to note that the article suggests a virus-mediated horizontal transfer of KCO3-like ancestor ion channels.  Overall article meets all qualities to be accepted for publication.

Author Response

Dear Editor and Reviewers,
We greatly appreciate your critical evaluation of our manuscript and helpful comments. Our reply to your comments would be provided point by point, where “A” stands for “Authors”, and “L” for “Lines”, where changes have been implemented. The language of the entire manuscript has been checked and corrected.

____________________________________________________________________________

The article have good collection of information on TPK ion channels, especially on its evolution. It is interesting to note that the article suggests a virus-mediated horizontal transfer of KCO3-like ancestor ion channels.  Overall article meets all qualities to be accepted for publication.

A: We thank you for your valuable opinion regarding our work.

Reviewer 3 Report

After carefully reviewing the paper, I have to say that it's rather interesting and covers a very original topic into plant TPK ion channels
The authors conducted an interesting study utilizing the application
of modern bioinformatic tools and a good research analysis method. The research shows, in any case, care and attention in the drafting of the text with quite original passages.

However: I suggest the authors to further extend and better illustrate the abstact
Passages repeated in the introduction! The conclusion part is to long and inadequate. It should be revised. I believe are in the submissive version, rather lacking and need more in-depth analysis in order to make the results and objectives of this article more clear. line 243
"Furthermore, new questions were raised during our study. To answer the questions and
explain the results of our study the several interconnected hypotheses"

Author Response

Dear Editor and Reviewers,
We greatly appreciate your critical evaluation of our manuscript and helpful comments. Our reply to your comments would be provided point by point, where “A” stands for “Authors”, and “L” for “Lines”, where changes have been implemented. The language of the entire manuscript has been checked and corrected.

____________________________________________________________________________

After carefully reviewing the paper, I have to say that it's rather interesting and covers a very original topic into plant TPK ion channels The authors conducted an interesting study utilizing the application
of modern bioinformatic tools and a good research analysis method. The research shows, in any case, care and attention in the drafting of the text with quite original passages.

A: We thank you for your valuable opinion regarding our work.

However: I suggest the authors to further extend and better illustrate the abstract

A: Thanks for your suggestion.  The abstract was further modified (L11-27). However, we wish to note that there is a limit (set by Instructions for Authors) of about 200 words. The original version had 204, and the modified – 219.

Passages repeated in the introduction!

A: The introduction section was modified and 1 reference was added additionally (please, see L56-59, 84-85)

The conclusion part is to long and inadequate. It should be revised.

A: We have modified the conclusion section according to suggestions of reviewer 3 (part was transferred to the Discussion, rest – revised) L496-505.

I believe are in the submissive version, rather lacking and need more in-depth analysis in order to make the results and objectives of this article more clear. line 243 "Furthermore, new questions were raised during our study. To answer the questions and explain the results of our study the several interconnected hypotheses"

A: The discussion section was further modified according to the suggestion of reviewer 3 (please, see L 244-275, where obtained results were discussed).

This manuscript is a resubmission of an earlier submission. The following is a list of the peer review reports and author responses from that submission.

Round 1

Reviewer 1 Report

This is a second resubmission; the language has been further edited and the paper reads quite well. Some limitations of the study have been addressed in the discussion; I believe this manuscript is acceptable in its present form.

Reviewer 2 Report

As clearly explained in my first review, my concerns were about Fig. 4. I might have erroneously referred to fig 3 in the second review, but the obvious mistake was and still is in Fig. 4. Fig. 4 is NOT a cladogram, it is a polygram with most relationships unresolved. The issue here is that the authors clearly have little familiarity with cladistics. To help the authors to see their mistake, I invite them to remove, for example, the four lineages between Clade II and the clade KCO3 proteins and place them at the top of the figure. You can do that because these four lineages independently diverge from the base. Done this, only good sense would prevent you from considering them as part of clade I. The same may apply to most other lineages. No scientist with a minimum experience in cladogenesis would consider Fig 4 a phylogenetic tree. The few clades resolved have too a low bootstrap to be reliable.

I suspect that the authors might have obtained more clear-cut results if they focused on a more limited topic instead of browsing through the whole extant diversity.

Just to have the view of a renowned expert in gene phylogeny, I suggest asking Professor Stefan A Rensing

Plant Biotechnology, Faculty of Biology, University of Freiburg, Germany, to see the paper. His e-mail address is:

[email protected]